# Structural Properties of Phenylalanine-Based Dimers Revealed Using IR Action Spectroscopy

**DOI:** 10.3390/molecules27072367

**Published:** 2022-04-06

**Authors:** Iuliia Stroganova, Sjors Bakels, Anouk M. Rijs

**Affiliations:** 1Division of BioAnalytical Chemistry, AIMMS Amsterdam Institute of Molecular and Life Sciences, Vrije Universiteit Amsterdam, De Boelelaan 1108, 1081 HV Amsterdam, The Netherlands; i.stroganova@vu.nl (I.S.); s.bakels@vu.nl (S.B.); 2Radboud University, FELIX Laboratory, Institute for Molecules and Materials, Toernooiveld 7, 6525 ED Nijmegen, The Netherlands

**Keywords:** infrared spectroscopy, self-assembly, molecular beam, peptide aggregates

## Abstract

Peptide segments with phenylalanine residues are commonly found in proteins that are related to neurodegenerative diseases. However, the self-assembly of phenylalanine-based peptides can be also functional. Peptides containing phenylalanine residues with different side caps, composition, and chemical alteration can form different types of nanostructures that find many applications in technology and medicine. Various studies have been performed in order to explain the remarkable stability of the resulting nanostructures. Here, we study the early stages of self-assembly of two phenylalanine derived peptides in the gas phase using IR action spectroscopy. Our focus lies on the identification of the key intra- and intermolecular interactions that govern the formation of the dimers. The far-IR region allowed us to distinguish between structural families and to assign the 2-(2-amino-2-phenylacetamido)-2-phenylacetic acid (PhgPhg) dimer to a very symmetric structure with two intermolecular hydrogen bonds and its aromatic rings folded away from the backbone. By comparison with the phenylalanine-based peptide cyclic *L*-phenylalanyl-*L*-phenylalanine (cyclo-FF), we found that the linear FF dimer likely adopts a less ordered structure. However, when one more phenylalanine residue is added (FFF), a more structurally organized dimer is formed with several intermolecular hydrogen bonds.

## 1. Introduction

The amino acid residue phenylalanine (F) is regularly present in peptide segments of proteins associated with neurodegenerative diseases. These phenylalanine-rich segments are prone to form amyloid fibrils [1,2,3]. However, this ability to form solid aggregates can be exploited when manufacturing nanomaterials and in biotechnology [4,5]. Depending on sample preparation conditions and chemical properties of the phenylalanine-based peptides, different nanostructures can be formed. The resulting nanostructures find applications in biomedicine and microelectronics, where they are used as optical waveguides, biocontainers, and optoelectronic nanomaterials [6,7,8,9,10,11]. 

In the work performed by Gazit et al. [12], the formation of nanotubes and spherical nanostructures in water of different phenylalanine-based peptides was demonstrated by TEM (transmission electron microscopy) and AFM (atomic force microscopy) techniques. Nanotubes were assembled from peptides consisting of two phenylalanine residues (*L*-phenylalanyl-*L*-phenylalanine, FF), while spherical nanometric assemblies were formed from a very similar peptide—namely, 2-(2-amino-2-phenylacetamido)-2-phenylacetic acid (PhgPhg). PhgPhg differs from FF in the aromatic side chain; PhgPhg lacks the methylene group between the Cα and the phenyl group. Both the FF nanotubes as well as PhgPhg nanospheres proved to be very stable physically and chemically. Tamamis et al. investigated the possibility of other phenylalanine-based peptides to form different types of nanostructures [13]. For example, the peptide *L*-phenylalanyl-*L*-phenylalanyl-*L*-phenylalanine (FFF) self-assembled into plate-like nanostructures in an aqueous environment. Using a vapor deposition procedure, Adler-Abramovich et al. showed that linear FF peptides can undergo cyclization and transform into cyclo-FF species, which self-assembled into nanorods [14]. Depending on solvents and preparation conditions, different types of nanostructures, such as nanofibers, microtubes, and microrods, can be formed from FF peptides [15]. Additionally, small alterations in side groups and charges—for example, in the case of the cationic dipeptide H-Phe-Phe-NH_2_·HCl—induce rearrangements of the formed nanotubes into vesicles upon dilution [16]. Depending on the used substrate, pH, and medium, changes in the morphology of FF nanotubes can be made [17].

In order to explain the remarkable stability of FF nanotubes, several theoretical and experimental studies were performed to unravel the properties of these nanostructures and the interactions that drive the self-assembly process [13,18,19,20,21]. Reches et al. showed that aromatic interactions, rather than electrostatic interactions between the charged termini, play a critical role in the self-assembly into tubular nanostructures of several FF-based peptides with different end caps [21]. Additionally, theoretical insights on the formation of various classes of nanostructures formed from FF and FFF peptides were provided by Tamamis et al., thereby highlighting different alignment in beta-strands [13]. The role of polar and electrostatic interactions in capped uncharged and uncapped zwitterionic FF peptides was discussed by the Scott Shell group [20]. Starting from the dimers, the uncapped peptides form so-called ordered “ladder-like” structures. The hydrophobic groups are organized in a stacked fashion to reduce their exposure to the solvent, and thus the backbones exposed to the solvent. The uncharged FF peptides formed less ordered aggregates, which are governed by polar interactions and stacking of the side chains. However, as Tamamis et al. discussed, it remained difficult to connect the observed structural large nano assemblies to the molecular-level small oligomeric system [13].

In this work, we focus on the structural characterization of proto-oligomers via the transformation of monomers into dimers. To reveal structural details and key intermolecular interactions upon dimerization, phenylalanine-based peptides were studied under isolated conditions in the gas phase, i.e., free from interactions with the solvent. There are a limited number of studies on gas-phase dimers of capped model peptides where key intra- and intermolecular interactions are examined [22,23,24]. An analysis of cyclo-FF clusters up to the tetramer has been recently published [25], as well as the structural characterization of other model dimers [26,27]. Monomers of capped Ac-FF-NH_2_ and Ac-FFF-NH_2_ peptides have been studied in the group of M. Mons [28,29]. Here, we present an experimental and theoretical study on the dimers of two different peptides—PhgPhg and FF—under isolated gas-phase conditions using a combination of IR-UV action spectroscopy and quantum chemical calculations. The choice of uncapped peptides was based on the fact that uncapped FF and PhgPhg peptides are able to form different types of nanostructures [12]. The current study focuses more on characteristic IR signatures of intra- and intermolecular interactions of the peptides themselves, unperturbed of solvent-induced interactions. The main goal of this work was to reveal the structural differences and similarities in the dimers of different phenylalanine-based peptides and to explore whether we can extrapolate these findings to the higher-order clusters and eventually the nanostructures that are formed in the condensed phase. However, it should be noted that our study focused on the gas phase, without a solvent environment where hydrophobic interactions are present that can contribute to the stability of oligomers.

## 2. Materials and Methods

### 2.1. Experimental Details

The experiments presented here were performed using a laser desorption molecular beam set-up. A detailed description has been published recently [30]; therefore, only a short explanation of the experimental methods is presented. The uncapped peptides 2-(2-amino-2-phenylacetamido)-2-phenylacetic acid (PhgPhg) and *L*-phenylalanyl-*L*-phenylalanine (FF) were purchased from BioMatik and Bachem, respectively, and used without any further purification. Both peptides consist of *L*-amino acids. The powdered peptides were mixed with carbon black in an approximate 50:50 ratio and were deposited on a graphite sample bar. The sample bar was placed in the source chamber in front of the nozzle of a pulsed valve (Jordan) on a movable stage to provide fresh sample every laser shot. The height of the sample bar with respect to the nozzle orifice was adjusted to have optimal dimer signal [27]. A pulsed Nd:YAG laser (New Wave Research, Polaris II, 1064 nm, 10 Hz) operating at a low pulse energy (1 mJ) was used to desorb the dipeptides from the sample bar. The desorbed molecules were subsequently cooled in a supersonic beam of argon with a backing pressure of 4 bar. About 10 cm downstream, the center of the molecular beam went through the skimmer and entered the laser interaction region. Here, the neutral molecules interacted with a UV beam produced by a UV laser (Nd:YAG pumped dye lasers, Liop-Tec, around 2.5 mJ/pulse, 10 Hz). The ions were produced by a one color (1 + 1) REMPI (resonance-enhanced multiphoton ionization) scheme [31] and were subsequently characterized using a reflector time-of-flight (TOF) mass spectrometer (Jordan) equipped with a dual microchannel plate detector (Jordan). For the IR-UV ion-dip spectroscopic experiments, the molecular beam was crossed perpendicularly by the UV laser and counter propagated by the IR laser [31]. The IR spectra were obtained using the Free Electron Laser FELIX [32] (pulse width 10 µs, pulse energies about 100 mJ) in the region 1800–100 cm^−1^ and an OPO/OPA system (LaserVision, pulse width 4 ns, pulse energies about 20 mJ) in the 3600–3000 cm^−1^ region. To record an IR spectrum, the UV wavelength was fixed at a resonant transition of a selected conformer, while the IR wavelength was scanned over the desired region. The IR laser arrived about 300 ns prior to the UV beam. When the IR frequency was resonant with a vibrational transition of the selected conformer, the population was transferred from the ground state to this vibrational level, resulting in a depletion of the ground state population, and thereby creating a dip in the ion signal. By measuring the ion yield of the selected mass while scanning the IR wavelength, a mass- and conformer-selected IR-UV ion dip spectrum was recorded. In order to correct for signal fluctuations, the IR laser was operated in an on/off manner. The IR laser was set at 5 Hz and the UV at 10 Hz. Every measured IR wavenumber was thereby an average of 30 shots with IR and 30 shots without IR. The resulting IR absorbance spectra were obtained by taking the logarithm of the ratio between on and off signals, divided by the power of FELIX at each wavelength and multiplied by the photon energy in wavenumbers. Every experimental spectrum was an average of multiple recorded spectra. Different conformers were identified using IR-UV hole burning spectroscopy. In this scheme, the IR laser was fixed on a vibrational transition of a single conformer. The UV laser was scanned, and depletion in the REMPI spectrum occurred for the peaks that belonged to the selected conformer.

### 2.2. Theoretical Approach

In order to assign structures to the studied peptides and their dimers, the experimental IR spectra were compared with theoretical spectra that were computed based on quantum chemical calculations. First, an amber force field was used to perform a conformational search [33]. Different starting structures were heated up to 1000 K for monomers, and up to 300 K (300 K, 275 K, 250 K, and 200 K) for dimers, and then cooled down to 0 K to investigate all the minima in the potential energy landscape. That search resulted in 500 structures, from which about 20 different unique structures could be derived. For the dimers, multiple conformational searches were performed in order to explore as many conformations as possible, while also multiple dimer structures were manually designed. Selected conformations, either from conformational search or manually designed, were optimized and their IR vibrations, along with the zero-point energy (ZPE)-corrected energies and Gibbs free energies at 300 K, were calculated using the B3LYP D3/6-311 + G(d,p) functional and basis set for the monomers and dimers. Since it is considered that barriers higher than 12–15 kJ/mol cannot be crossed during supersonic cooling, we assume that the conformational population at 300 K, when molecules are desorbed, is preserved at lower temperatures [34,35,36]. The computations were performed in the Gaussian 16 environment [37]. In the IR region 1800–100 cm^−1^, a scaling factor of 0.976 was used to correct for anharmonicity, while in the amide A region 3600–3000 cm^−1^ a scaling factor of 0.956 was applied.

## 3. Results

We describe the structural assignments and IR spectroscopic features of two uncapped phenylalanine-based peptides—namely, PhgPhg and FF; see Figure 1a and Figure 3a, respectively. PhgPhg consists of two phenylglycine residues (Phg) and FF of two phenylalanine residues (F) connected via a peptide bond. Both peptides are very similar except for the CH_2_ group. The extra CH_2_ moiety that connects the aromatic rings to the Cα in the FF dipeptide makes it more flexible than the PhgPhg dipeptide.

### 3.1. Structural Assignment of the PhgPhg Monomer

As can be seen in Appendix A, the UV excitation spectrum shows the presence of a large progression of sharp, well-resolved peaks, which originate from one dominant conformation (conformer A). A second minor conformation was found based on IR-UV hole-burning experiments (conformer B in this work; see Appendix A). The IR experimental spectra of both conformations of the PhgPhg monomer are presented in Figure 1b,c, respectively (black traces). They were recorded at a UV excitation wavelength of 37,426 cm^−1^ for conformer A and 37,319 cm^−1^ for conformer B. 

The quantum chemical calculations for the PhgPhg monomer resulted in multiple structural arrangements that could be combined in families. The structures were grouped into four families based on their characteristic intramolecular hydrogen bond patterns, similar to the previously reported study on dipeptides [38]. These are presented in Figure 1a with their zero-point and Gibbs free energies in kJ/mol between brackets. The conformations shown here represent the lowest energy structures of each family based on their hydrogen bond pattern. The conformers in family I are characterized by the free (not hydrogen-bonded) OH group and two C5 interactions. The most stable structure (Ia) has a bifurcated hydrogen bond, where the NH moiety interacts with both the C=O and NH_2_ groups. The benzene rings are roughly T-shaped at a small angle to each other. The Ib conformer, although structurally similar to Ia, has a different orientation of the COOH group, where the NH is hydrogen-bonded to the carboxylic OH moiety. The II family contains a strong hydrogen bond between the OH and the peptide C=O groups. The IIa conformer has a hydrogen-bonded NH group and the IIb conformer a free NH group, resulting in a higher energy. The III family can be classified by its free OH group, similar to the I family. However, the aromatic rings are not interacting with each other due to a rotation around the Cα-CO bond. IIIa is the most stable structure in this family and has two hydrogen bonds in a C5 configuration. IIIb is higher in energy and has only one hydrogen bond (C5). Similarly, like IIIa, IIIc has two C5 interactions; however, here the NH group interacts with the free electron pair of the carboxylic C-OH group. The IV family, which has a much higher energy, is also characterized by a hydrogen-bonded OH group, but unlike the II family, here the hydrogen bond is formed with the NH moiety.

#### 3.1.1. Conformer A of the PhgPhg Monomer

The experimental IR spectrum can roughly be divided in three regions: the amide A region (3600–3300 cm^−1^), the amide I and II plus fingerprint region (1800–1000 cm^−1^), and the far-IR regime (below 1000 cm^−1^). Focusing on the amide A region, an initial structural assignment can be made based on the NH and OH stretch vibrations. Here, the experimental IR spectrum of conformer A shows two dominant peaks—namely, at 3415 cm^−1^ (NH stretch) and 3583 cm^−1^ (OH stretch); see Figure 1b (black trace). This NH stretch frequency is typical for a weakly bonded NH group—for example, in a C5 interaction or with the π-cloud of an aromatic ring [26,39]. The position of the OH stretch peak corresponds to a free OH group. As can be seen from Appendix A, which shows the comparison of the experimental and calculated IR spectra of the lowest energy structure of each family, only structures from the I family show a good agreement. The structures from the II and IV families have a hydrogen-bonded OH group and therefore do not predict the observed free OH stretch peak at 3583 cm^−1^ well. The structures from the III family show a good agreement with the experimental OH vibration; however, the NH stretching peak is too much blueshifted. The lowest energy structure Ia shows the best agreement based on peak position and peak spacing, although the tiny peak at 3427 cm^−1^ was absent in our experimental IR spectrum as a result of experimental sensitivity limitations. Structure Ib from this family also shows good agreement in the amide A region; however, as described below, the most stable structure Ia shows a much better agreement with the experiment in the mid-IR and especially the far-IR region.

The mid-IR and predominantly the far-IR region (1800–100 cm^−1^) provides more details about both local as global structural characteristics and allows for a more precise structural assignment. The IR spectrum shows two peaks in the amide I region—one resulting from the carboxylic C=O group (1783 cm^−1^) and the other one originating from a peptide bond C=O (1717 cm^−1^). The calculated vibrations of all the families are summarized in Appendix A. Based on the comparison of theoretical frequencies with experimental values, the position in the experimental spectrum of the carboxylic C=O is typical for a free carboxylic group. The frequency of the peptide C=O vibration suggests that this C=O is also free. The region between 1650 and 1550 cm^−1^ shows a small peak at 1600 cm^−1^ that originates from an NH_2_ scissoring vibration. Another characteristic peak is the NH bend vibration at about 1500 cm^−1^, originating from either hydrogen-bonded C5 interactions or π-interactions with aromatic rings. Coinciding with the 3 μm region, also the amide I and II peak positions and shifts are well represented by conformer Ia (see Figure 1b, top panel).

The far-IR region below 500 cm^−1^ comprises many well-resolved peaks that correspond to IR transitions mainly originating from global motions where the entire molecule is involved in vibrations. This region proves to be very diagnostic for distinguishing between the structures within one structural family. All far-IR peaks are very well represented by the calculated spectrum of structure Ia, such as the quartet of peaks between 450 and 510 cm^−1^, the doublet at 372 and 360 cm^−1^, and multiple peaks below 300 cm^−1^. The only difference in this region between the experimental and assigned spectrum of Ia is the presence of peaks around 300 cm^−1^. Here, the experimental spectrum shows a doublet at 293 cm^−1^ and 300 cm^−1^, while the calculated Ia structure has a single intense peak at 313 cm^−1^ (NH_2_ rocking) and a minor, redshifted peak at 294 cm^−1^ (global motion involving the NH_2_ group). However, based on the entire IR region, we confidently assign conformer A to the most stable Ia structure.

#### 3.1.2. Conformer B of the PhgPhg Monomer

The experimental IR spectrum of conformer B shows a distinctive peak at 3583 cm^−1^ and a doublet at 3426 cm^−1^ and 3414 cm^−1^ in the amide A region; see Figure 1c (black trace). The region between the doublet and the single peak does not show any IR activity; see Appendix A. Note that the spectrum in Appendix A is very noisy, resulting from the low number of averages. The assignment of conformer B follows the same workflow as for conformer A, starting with the analysis of the 3 μm region, followed by the amide I and II regime, and finally the far-IR part of the spectrum. The peak at 3582 cm^−1^ corresponds to the free OH group and is similar to the respective vibration of conformer A (3583 cm^−1^). The IR region around 3400 cm^−1^ is however different for conformer B, where two peaks are observed. The frequency of 3414 cm^−1^ coincides with the peak observed for conformer A (3415 cm^−1^); however, the IR spectrum of conformer B shows an additional peak at 3426 cm^−1^. The first peak at 3414 cm^−1^ corresponds to the NH_2_-stretching, which has a lower intensity than the NH stretching peak. The second peak might originate from a weakly bonded NH stretch mode—for example, a C5 interaction. Therefore, it can be assumed that the structure for the second conformer should have a free OH and a weakly bonded NH group, which corresponds to the III family. The calculated IR spectra originating from this III family are presented in Appendix A. The lowest structure from this family—namely, IIIa—shows an adequate agreement with experiment; see Figure 1c. However, the spacing between the two peaks around 3400 cm^−1^ is predicted to be smaller in the calculations than observed in our experimental data. 

This assignment is further substantiated by analyzing the IR spectra in the mid- and far-IR region (1800–100 cm^−1^). The experimental spectrum in the amide I region consists of two peaks—namely, a peak at 1770 cm^−1^ and one at 1700 cm^−1^. The former originates from the carboxylic C=O group, while the latter results from a peptide bond C=O group. Both peaks are slightly redshifted compared with the corresponding peaks in conformer A, which indicates that both C=O groups of conformer B are hydrogen-bonded. The next peak in the spectrum is located at about 1601 cm^−1^ and corresponds to an NH_2_ scissoring vibration. The NH bend vibration at 1495 cm^−1^ in the amide II region is also slightly redshifted compared with the respective mode of conformation A. This can indicate that the NH group is involved in a weaker hydrogen bond for conformer B. All these findings support the hypothesis that the structure of conformer B belongs to the III family. Based on the comparison with theoretical frequencies presented in Appendix A and the IR spectra of the families shown in Appendix A, we can exclude IIIc. The IIIc conformer has a free carboxylic C=O group, which results in a corresponding blueshifted C=O transition compared with the experimental signature. Both IIIa and IIIb, which differ mainly in the position of the amide NH_2_, agree well with the experiment in the amide I and II region. However, the NH_2_ scissoring mode at about 1601 cm^−1^ is much better predicted for the IIIa conformer. This is because in IIIa the amide group interacts with the peptide C=O moiety, while in IIIb the NH_2_ is free. 

The far-IR region, especially below 500 cm^−1^, proves to be very useful for structural analysis. The calculated spectrum of structure IIIa shows an almost perfect agreement with the experimental spectrum of conformer B. In detail, a good agreement between the spectra is observed for the single peak at 505 cm^−1^, the doublet at about 475 cm^−1^, the single peak at 404 cm^−1^, the quartet of peaks between 360 and 300 cm^−1^, and between 240 and 180 cm^−1^. The only difference between the experimental and assigned spectra is, similar with conformer A, the presence of the two peaks around 300 cm^−1^. The experimental spectrum has a doublet here at 278 cm^−1^ and 290 cm^−1^, while the calculated IIIa structure shows a single intense peak at 282 cm^−1^ (NH_2_ rocking). However, based on the whole IR region, we confidently assign conformer B to the most stable structure from the III family—IIIa, which is also the overall lowest energy structure at 300 K.

### 3.2. Structural Assignment of the PhgPhg Dimer

The REMPI spectrum of the PhgPhg dimer is shown in Appendix A. It has a broad feature around 37,600 cm^−1^ (origin) and another more intense broad band with its maximum around 38,000 cm^−1^. The latter peak was previously assigned for the single phenylglycine monomer to the vibrational band of an out-of-plane motion of OH with respect to the COOH plane [40]. To obtain the IR spectrum of the PhgPhg dimer using IR-UV ion dip spectroscopy, the UV laser was set at the maxima of the peaks—namely, at 37,603 cm^−1^ and 37,988 cm^−1^. This resulted in identical IR spectra, indicating that the dimer exists as one dominant conformer under our experimental conditions; however, considering the broad UV spectrum, we cannot rule out the presence of other conformations.

The experimental IR spectrum of the PhgPhg dimer is shown in Figure 2a (bottom, black trace). The amide A region (3300–3650 cm^−1^) gives general information about the structure of the dimer. First, there are no peaks present around 3600 cm^−1^ (typical for a free OH stretch signature), which means that both OH groups are hydrogen-bonded in the dimer. The region around 3400 cm^−1^ (NH stretch) comprises three peaks—namely, at 3338, 3369, and 3411 cm^−1^. The most intense peak at 3369 cm^−1^ originates from an NH stretch mode, while the other two lower intensity peaks are coming from NH_2_ stretching vibrations. The region around 3000–3200 cm^−1^ shows three distinctive peaks at 3045, 3076, and 3098 cm^−1^, which are located on top of a broad feature. The IR vibrations here around 3000 cm^−1^ originate from CH and CH_2_ stretches, which have typically limited diagnostic value for assignment to different types of backbones [41]. To further investigate the origin of the three peaks observed between 3000 and 3100 cm^−1^, we compare the experimental spectrum with those of conformation A of the PhgPhg monomer and with the cyclo-FF monomer [25]; see Appendix A. As can be seen, the same pattern of peaks is present in all three spectra. The common feature amongst these three molecules is the aromatic side groups. Therefore, it can be assumed that the three peaks originate from CH stretching vibrations in these phenyl groups. For the PhgPhg dimer, these three peaks are located on top of a broad feature which ranges from 2950 to 3200 cm^−1^. In addition to CH stretching vibrations, also hydrogen-bonded OH stretch vibrations can be found here. The OH stretch vibration experiences a significant redshift and broadening when hydrogen-bonded. It should be noted that the cyclo-FF monomer does not have an OH group at all, while in the PhgPhg monomer the OH is free. The broad band of the PhgPhg dimer is therefore assigned to strongly hydrogen-bonded OH stretching vibrations.

The characteristic theoretical vibrations of C=O and NH groups for the dimer of PhgPhg are summarized in Appendix A. They are obtained from theoretical calculations of different types of PhgPhg dimers. The carboxylic C=O stretch is observed in the experimental spectrum at 1748 cm^−1^ with a shoulder on the red side of the peak. Based on the comparison with theoretical values, it can be assumed that both carboxylic C=O groups are most likely not involved in (intermolecular) hydrogen bonds, while the carboxylic OH moieties are hydrogen-bonded. The experimental peptide C=O vibration is located at 1660 cm^−1^ with a shoulder on the red side. This suggests that both peptide C=O groups are involved in a hydrogen bond. The broad peak in the spectrum at about 1607 cm^−1^ corresponds to an NH_2_ scissoring vibration. This vibration does not correspond well to the theoretical values (see Appendix A); however, it can be assumed that the amide groups are either involved in intramolecular interactions with NH as in conformer A of the monomer or with a peptide C=O as in conformer B. The amide II region consists of a broad peak corresponding to the NH bending vibration, with the maximum located at about 1522 cm^−1^. Although, this transition is slightly blueshifted compared with the corresponding peak in the conformers of the PhgPhg monomer, the calculated frequencies in Appendix A indicate that the NH groups are not involved in strong intermolecular interactions. The intramolecular hydrogen bonds are retained within the monomeric units.

Previous experiments have shown that often the monomeric conformations are conserved in the dimer structure [25,26]. However, there are also counterexamples on adaptive aggregation of acetylated glycine, alanine, and dialanine esters where the structure of the monomer is transformed from a stretched to a more folded conformation upon dimerization [42]. Based on the analysis of the IR frequencies in the amide A, I, and II region, it seems that this also holds for the PhgPhg dimer. Still, the IR spectrum of the dimer is different from the spectra of both monomeric conformations. The involvement of the OH group and peptide C=O in intermolecular hydrogen bonds in the dimer can be seen directly from the IR spectrum, where both peptide and carboxylic C=O stretching peaks are redshifted compared with the respective transitions of both conformers of the monomer. Conformer B is not observed in the dimer, as the two phenyl groups point to opposite sides of the peptide backbone, thereby hindering the direct formation of intermolecular hydrogen bonds.

In order to make structural assignments on the PhgPhg dimer, an extensive conformational search was performed based on simulated annealing. In addition, multiple structures were manually designed from combining two times the monomer of conformer A or conformer B, or a mixture of conformer A and B, after which their IR frequencies were calculated. The resulting 136 different structures were grouped into three structural families based on the orientation of the monomers with respect to each other: parallel, anti-parallel, or random orientation of the N- and C termini of the monomer units in the dimer. Within the defined families, dimers with different intermolecular hydrogen bonds were highlighted. The lowest energetic structures almost exclusively correspond to anti-parallel structures, while the first parallel structure was over 20 kJ/mol higher in energy than the lowest energy structure. This allowed us to exclude most parallel structures. Seven of the lowest energy structures from the anti-parallel 2×CO-OH family are plotted in Appendix A for comparison, along with the most stable dimer (c_min). The structures that belong to this anti-parallel 2×CO-OH family have two intermolecular C=O⋯OH hydrogen bonds between the carboxylic OH and the peptide C=O groups, while c_min dimer is an anti-parallel structure with C=O⋯OH and OH⋯NH_2_ intermolecular hydrogen bonds. 

The transitions in the amide A region around 3400 cm^−1^ of the c_min structure show quite a good agreement with the experiment. The OH stretch vibration at 3075 cm^−1^ is located at approximately the middle of the broad band of the experimental spectrum. The strongly redshifted peak of the OH stretch vibration at 2800 cm^−1^ that is present in the c_min dimer originates from the OH⋯NH_2_ hydrogen bond. However, based on the comparison with the experimental spectrum in the amide I region, the c_min dimer can be excluded. One of the peptide C=O groups is too far blueshifted to 1700 cm^−1^, as it is not involved in an intermolecular hydrogen bond between the two monomers. The lowest energy structure from the 2×CO-OH family c1 (Figure 2a) shows a good agreement with the experimental spectrum in the amide A region. Only the position of the calculated strong OH stretch vibration is not exactly centered at the experimentally observed broad peak but was calculated slightly redshifted. The NH and NH_2_ stretching vibrations around 3400 cm^−1^ match best to the theoretical values of the lowest energy c1 dimer, considering the other structures of the 2×CO-OH family. The mid-IR region (1800–1000 cm^−1^) is not diagnostic; all calculated spectra of 2×CO-OH family are rather similar in this region and have a good overlap with the experiment. However, the far-IR part of the spectrum proves to be highly diagnostic. It comprises an intense peak at 512 cm^−1^, followed by a broad feature at 462 cm^−1^, a gap with no peaks between 450 and 415 cm^−1^, a smaller peak at about 400 cm^−1^, and a strong vibration at 342 cm^−1^. These transitions are best predicted in the c1 structure. Therefore, the PhgPhg dimer is assigned to the lowest energy structure c1 from the 2×CO-OH family. Figure 2b presents the 3D representation of this structure. This c1 dimer is a centro-symmetric molecule that consists of two similar monomeric units that mainly retain the structure of the conformation A. However, the orientation of the carboxylic group of both monomers is slightly altered in order to ensure the formation of intermolecular C=O⋯OH hydrogen bonds. 

### 3.3. Structural Assignment of the FF Monomer

The electronic and vibrational spectra of the FF monomer, as well as the structural assignment based on the amide A region, were reported previously by Abo-Riziq et al. [43] and Pérez-Mellor et al. [44]. In the experiments presented in this work, the IR region was extended to the mid- and far-IR. The REMPI and hole-burning spectra of the monomer and dimer can be found in the Appendix A. The presence of two conformations was confirmed by IR-UV hole-burning experiments (see Appendix A). The IR spectrum, presented in Figure 3a, was recorded at a UV wavelength of 37,508 and 37,532 cm^−1^ for conformer A and of 37,544 cm^−1^ for conformer B. The structural families, the details of our structural assignment, and the corresponding full IR spectra are shown in Appendix A. Therefore, only a short overview of the assigned structures and the IR spectra of the two conformations of the FF monomer in the region 1800–100 cm^−1^ are presented here.

Both conformations are assigned to structures with a bifurcated hydrogen bond, where the NH group is hydrogen-bonded to both the carboxylic C=O and the N-terminal NH_2_ group (see Figure 3a). Conformer A is assigned to the most stable structure that has two aromatic rings T-stacked. The structural assignments for conformer A and B that were reported previously by Abo-Riziq et al. [43] are the same as those in our results; however, the structures are assigned vice versa (see details in Section 7 of Appendix A). In the work of Pérez-Mellor et al. [44], the assignment for conformer A is exactly the same as in our experiments. Therefore, despite the difference in functionals and basis sets that were used previously and in our work, we confirm their structural assignment for conformer A, which is substantiated by the mid- and far-IR spectrum (see Figure 3b).

We assign conformer B to the structure from the same conformational family (I) as conformer A, which is characterized by a bifurcated hydrogen bond involving the NH group (see Figure 3a). However, this structure (Ia′) is more extended due to one of the aromatic rings facing outwards. This results in an increase in energy of 10.5 kJ/mol with respect to the lowest energy structure. The structural assignment of conformer B made by Pérez-Mellor et al. is similar to our result, except for the rotated carboxyl (COOH) group. An extended comparison with previously reported assignments is shown in Appendix A. Based on the mid-IR region, especially the peaks between 1400 and 950 cm^−1^, we make the structural assignment of conformer B to the Ia′ structure (see Figure 3b).

### 3.4. Structural Assignment of the FF Dimer 

The REMPI spectrum of the dimer of FF is relatively broad and is shown in the Appendix A. The IR spectra were measured at several different UV excitation wavelengths, which resulted in the same IR spectra of the FF dimer. However, considering the broadness of both UV and IR spectra, we cannot exclude the presence of multiple conformers for the FF dimer. The IR spectrum of the dimer of FF is shown in Figure 3b (black, bottom trace). The spectrum in the 3 μm region was also recorded, but no reliable peaks were observed due to a low ion signal of the dimer and a low signal-to-noise ratio (not shown). The IR spectrum in the mid- and far-IR region comprises several relatively broad peaks compared with the FF monomer—especially the far-IR regime lacks many well-resolved transitions. The overall broadness of the IR spectrum can be attributed to the insufficient cooling in the molecular beam. That can be due to the flexible nature of the dimer originating from limited (intermolecular) interactions and the flexibility of the aromatic rings in both FF monomer and dimer compared with cyclo-FF [25] and the above-discussed PhgPhg.

The first peak in the IR spectrum of the dimer at 1747 cm^−1^ originates from both carboxylic C=O stretch vibrations. It is broadened and redshifted compared with the monomer, indicating that the environment of carboxylic groups in the dimer has altered. Based on the comparison with theoretical frequencies from calculations, which are shown in Appendix A, the carboxylic C=O can be either free with the neighboring OH moiety involved in an intermolecular hydrogen bond or it forms an intermolecular hydrogen bond with the NH group. The peaks of the peptide C=O stretch vibrations are located at 1691 and 1654 cm^−1^. The first peptide C=O group (1691 cm^−1^) is not involved in intermolecular interactions, as it can be found at the same position as the corresponding transition of conformations A and B of the FF monomer. The peak at 1654 cm^−1^ is redshifted compared with the corresponding peak of the conformations of the FF monomer, indicating that this peptide C=O group is involved in an intermolecular hydrogen bond. This is in agreement with the theoretical values presented in Appendix A. 

The amide II region in the spectrum of the FF dimer shows a broad peak around 1614 cm^−1^ with a shoulder on the red side resulting from the NH_2_ scissoring vibration. The lone pair of the NH_2_ group in both conformers of the FF monomer is connected with the NH group, and its vibration is predicted at 1625 cm^−1^. Based on the comparison with theoretical frequencies shown in Appendix A, it can be assumed that both NH_2_ groups can have an intermolecular hydrogen bond with the C=O or NH groups. However, since the NH and C=O transitions, when bonded to NH_2_ group, do not correspond well to the experiment, we assume that NH_2_ is not involved in intermolecular hydrogen bonds. The following peak in the spectrum is NH bending vibration at 1522 cm^−1^. It overlaps well with the experimental IR spectrum of the PhgPhg dimer where both NH moieties are not involved in intermolecular hydrogen bonds. According to the theoretical values listed in Appendix A, the NH groups in the FF dimer are not involved in intermolecular hydrogen bonds. These findings indicate that only one intermolecular hydrogen bond is formed, in contrast with the PhgPhg dimer, which has two intermolecular hydrogen bonds. In the FF dimer, the peptide C=O of one of the monomers is expected to be hydrogen-bonded to the OH of the carboxyl group of the other monomer. Both NH groups are not involved in any intermolecular hydrogen bonding in the dimer. The intramolecular hydrogen bonds of the FF monomer are expected to be preserved in the FF dimer. Since both conformer A and B of the FF monomer have the same intramolecular hydrogen bonds, the FF dimer can be formed from both conformers.

In order to assign the structure of the dimer, an extensive conformational search was performed starting from different combinations of the assigned monomers in various dimer configurations. In addition, a number of dimers were manually designed, and their IR transitions were calculated. That resulted in 116 unique structures that were combined into families based on their hydrogen bond pattern and orientation of the monomers within the dimer, in a similar fashion as for the PhgPhg dimers. There was no preference in energy over (anti-) parallel structures. However, dimers that were formed by two conformers B of the monomer were typically 19 kJ/mol higher in energy compared with the most stable dimer based on monomer A conformations. The structures with energies above 35 kJ/mol were excluded from further analysis. The calculated IR spectra of the remaining 80 structures were compared with the experimental IR spectrum; however, none of these structures showed a perfect agreement with the experiment. Nineteen structures that represent different intermolecular hydrogen bond patterns are shown in the Appendix A. Based on our experimental findings, we expect in the FF dimer one intermolecular hydrogen bond between peptide C=O and OH moiety (see Figure 3a, bottom). Although, from our 80 structures, no calculated IR spectrum matched convincingly when considering the amide I and II region. The possible structure is expected to have an unchanged intramolecular hydrogen bond pattern of the FF monomers. Since dimers consisting of two conformer B of the FF monomers were high in energy, the FF dimer is likely formed from either conformer A or from both conformer A and B.

## 4. Discussion

The main conformers A of both PhgPhg and FF monomers follow the same hydrogen bond pattern with NH⋯C=O and NH⋯NH_2_ intramolecular hydrogen bonds. These dominant species in our experiments are for both molecules the most stable structures found in the conformational search. Conformation A of the FF monomer has its aromatic rings T-stacked, while in the case of the main conformer of PhgPhg, the phenyl residues are orientated at an angle with respect to each other. Aromatic residues interacting with each other and being folded away from the backbone for these main conformers A allow the formation of intermolecular hydrogen bonds upon dimerization. However, conformers B of FF and PhgPhg are different from each other. In conformer B of PhgPhg, the aromatic rings are rotated around the Cα-CO bond and therefore not interacting with each other. This results in two intramolecular hydrogen bonds, i.e., C5 interactions in the form of NH⋯C=O and NH⋯NH_2_, at opposite sides of the peptide backbone. The position of the chromophores, pointing to the different sides of the backbone, makes the hydrogen bonding sites sterically hindered and less accessible than in conformer A for dimer formation. Accordingly, conformer B of PhgPhg was not observed as a building block for the dimer. Conformer B of FF has a similar backbone as the main conformer A, but the aromatic ring near the N-terminus is rotated away from the backbone, which results in a more extended structure and increase in energy compared with the main conformer A. Interestingly, other types of dipeptides—namely, Trp-Gly and Gly-Trp—also have similar unfolded backbones [38]. Main conformers of Trp-Gly and Gly-Trp possess the same two C5 interactions as conformers A of PhgPhg and FF. However, the second conformer of Gly-Trp has a strong C=O⋯OH intramolecular hydrogen bond that was not observed in the current study. As was stated in the previous work of the Mons group on capped FF peptides, aromatic interactions play an important role in the molecular conformation of aromatic peptides [28,29]. We observed similar aromatic interactions for conformer A of the FF peptide as in the main conformer of the Ac-FF-NH_2_ peptide. In case of the main conformer A of the PhgPhg monomer, the aromatic residues are also interacting; however, the minor conformers of both PhgPhg and FF peptides do not have these interactions.

The dimer of PhgPhg is assigned to an anti-parallel symmetric structure with two intermolecular C=O⋯OH hydrogen bonds, while the aromatic rings are pointing to the outside of the molecule. Two intermolecular hydrogen bonds are formed between the peptide C=O group and OH group from the C-terminus, which are the only moieties that are not involved in intramolecular bonds within conformer A of the monomer. Therefore, the structure of conformer A of the monomer is generally retained in the assigned dimer. However, the carboxylic group of the monomers is slightly altered to allow intermolecular interactions. The finding that the monomeric structure is retained in the dimer corresponds well to previous studies of peptide aggregation in our group.

For the FF dimer, one peptide C=O is likely free, and the other is hydrogen-bonded, which is different compared with the PhgPhg dimer, where both peptide C=O groups are hydrogen-bonded. This can be seen in the IR spectra of the dimers in the amide I region (see Figure 4), where the dimer of FF has two peaks between 1600 and 1700 cm^−1^, while the PhgPhg dimer has only one peak. The other peaks in the amide I and II range corresponding to carboxylic C=O stretch and NH bending vibrations are quite similar for both FF and PhgPhg dimers. The assignment for the FF dimer is most likely a structure with one C=O⋯OH intermolecular hydrogen bond (see Figure 3a, bottom). The FF dimer is formed from two monomers with conformation A or from a combination of conformer A and B. The monomeric structure is expected to be retained in the FF dimer, as was observed for the PhgPhg dimer and previously for other model peptides [26]. The far-IR spectrum of the FF dimer comprises two peaks at 486 cm^−1^ (global motion) and 340 cm^−1^ (NH_2_ rocking vibration). This pattern is very similar to conformer B of the FF monomer. Therefore, the presence of conformer B is expected in the FF dimer.

The IR spectra of PhgPhg and FF dimers are compared with two other phenylalanine-based peptides that form different types of nanostructures [12]—namely, uncapped *L*-phenylalanyl-*L*-phenylalanyl-*L*-phenylalanine (FFF) and 3,6-diphenylpiperazine-2,5-dione (cyclo-FF). Uncapped FFF is a peptide of three phenylalanine residues, and cyclo-FF is formed from an FF peptide by connecting the N- and C- termini together in a diketopiperazine ring and water loss. The REMPI spectrum of the FFF dimer is shown in Appendix A. The IR spectra of the FFF dimer and cyclo-FF dimer are shown in Figure 4 (blue and green traces, respectively), together with the dimer spectra of PhgPhg and FF. Details of the structural assignment of cyclo-FF can be found in a recent publication from Bakels et al. [25]. The IR spectrum of the cyclo-FF dimer in the amide I and II region has a significantly different IR pattern than the rest of the dimers, since the cyclo-FF monomer represents a cyclic molecule. Additionally, its far-IR spectrum below 500 cm^−1^ (mostly global motions) is rather different from the rest of the dimers—for example, it is lacking the NH_2_ rocking vibration since this group is not present in cyclo-FF. In the cyclo-FF dimer, two intermolecular hydrogen bonds are formed between the NH and C=O groups from the diketopiperazine ring, while in the FF and PhgPhg dimers, the intermolecular hydrogen bonds involve peptide C=O and carboxylic OH groups. In the case of the PhgPhg dimer, an antiparallel dimer is formed, where the N and C-termini of the monomers are orientated antiparallel to each other. However, no β-sheet type structures, with a typical C=O⋯NH hydrogen bond pattern, are formed for the FF, PhgPhg, and cyclo-FF dimers. 

Theoretical calculations for the FFF dimer are more complicated than for dipeptides due to the larger size of the molecule and more flexible nature of the backbone. However, the possible hydrogen bond pattern in the dimer could also be predicted by comparing the IR spectrum of the FFF dimer with the other peptides. The FFF dimer generally shows similar spectral features as the FF dimer; see Figure 4. However, the IR bands in the FFF dimer are less broadened compared with the FF dimer. The position of the carboxylic C=O vibrations suggests that one carboxylic C=O has an intermolecular hydrogen bond, probably with an NH group, while the other C=O can be free, but the OH moiety is then involved in hydrogen bonds, probably with the peptide C=O. The peak at 1697 cm^−1^ corresponds to free peptide C=O groups. The second peak at 1650 cm^−1^ indicates the presence of hydrogen-bonded C=O groups due to the redshift. The amide II region around 1500 cm^−1^ consist of two peaks, which correspond to hydrogen-bonded (blueshifted) and free (redshifted) NH groups. The far-IR spectrum of the FFF dimer is more similar to the FF dimer than to the PhgPhg dimer. In conclusion, the FFF dimer is expected to have more ordered structure than the FF dimer. The possible structure of the FFF dimer is shown in Appendix A. Three intermolecular hydrogen bonds are probably present in the FFF dimer. One hydrogen bond is formed between the peptide C=O and carboxyl group OH, while the NH groups likely form two intermolecular hydrogen bonds with a peptide and a carboxylic C=O groups.

It would be interesting to compare our findings with previously reported model dimers with aromatic content that were studied in the gas phase in order to elucidate β-sheet signatures. For example, in case of the Ac-Phe-OMe peptide with one amide bond, a C10 intermolecular hydrogen bond pattern was formed [22]. In the case of the PhgPhg dimer, we also observed the formation of a 10-membered ring. However, since we studied uncapped peptides, intermolecular bonds were formed not between C=O and NH groups but between C=O and OH groups at the C-termini. Another dimer of the Ac-Phe-NHMe peptide with two amide groups formed a 14-membered ring [45], which we did not observe in our study due to the lack of the second amide group in our uncapped peptides. The FF dimer seems to have only one intermolecular hydrogen bond that is not related to β-sheet structures; however, the presence of multiple conformations cannot be excluded. The FFF peptide contains two amide groups, which are required for the formation of a complete β-sheet unit cell. The possible structural assignment of the FFF dimer represents an antiparallel β-sheet with C10 and C14 intermolecular rings. This is in excellent agreement with the previously reported Ac-Val-Tyr(Me)-NHMe dimer that was the first reported β-sheet structure formed in the gas phase without solvent interactions or peptidic environment [23]. The FFF dimer represents additional experimental evidence of formation of fundamental β-sheet structure.

Different IR signatures in all of the above-discussed dimers could potentially lead to the formation of various nanostructures in the condensed phase. For example, the PhgPhg dimer can be a nice template for further aromatic stacking interactions in aggregation since it has aromatic side chains oriented to the outside of the backbone. Indeed, there are multiple computational and experimental publications suggesting that the aromatic side chains adopt the *syn* orientation in FF aggregates (θ = 0°, i.e., the side chains are located on one side of the peptide backbone) [46,47,48]. However, a theoretical study on the conformational dependence of capped FF peptides pointed out possible conformational transitions involving peptides with large side chain angles (θ = 180°, *anti* configuration) during the self-assembly process [49]. The existence of minor conformer B of PhgPhg (*anti* configuration) that was not observed in the PhgPhg dimer could possibly contribute at a later stage of aggregation to the closure and transition of double-layered structures into spherical assemblies.

The FF dimer is expected to be flexible with only one intermolecular hydrogen bond. In the theoretical study conducted by the group of Scott Shell, they also predicted the formation of less-ordered oligomers in case of uncharged FF dimers [20]. In the early work of Görbitz, the uncapped FF peptides formed hollow nanotubes that consist of a characteristic pore of six dipeptides oriented in a hexagon manner [18]. However, due to the charged termini, head-to-tail interactions of peptide backbones were present that are not relevant in our study of neutral peptides. Interestingly, the monomeric structure of FF they reported is very similar to the conformer A of FF monomer reported in our study, where the peptide backbone is linear and two aromatic residues have the *syn* orientation. Another study on the protected uncharged Ac-FF-NH_2_ peptide by Chakraborty et al. reported a completely different molecular arrangement, where the two aromatic side chains are pointing to opposite sides of the peptide backbone and the overall structure is stabilized by three intermolecular hydrogen bonds, resulting in a parallel β-sheet [50]. The uncapped FF dimer that we studied does not have enough hydrogen bond donors and acceptors to allow the same structure to form. However, in the case of the FFF dimer, a formation of anti-parallel β-sheet unit cell is predicted. Computational simulations on FFF aggregates by Tamamis et al. [13] show the formation of an anti-parallel β-sheet for the FFF dimer with hydrogen-bonding interactions between the internal peptide groups and between the charged uncapped termini. We also predicted an anti-parallel β-sheet type structure for the FFF dimer; however, interactions between terminal groups were absent in our experiments as a result of the neutral nature of our peptides. A follow up study of phenylalanine-based peptides with charged termini using ion mobility-mass spectrometry could provide complementary insights into the early stage of peptide self-assembly in solution [51].

## 5. Conclusions

Using IR-UV mass- and conformer-selective IR spectroscopy together with a quantum chemical theoretical approach, structures of two very similar phenylalanine-based dipeptides PhgPhg and FF and their dimers were determined. The far-IR region proves to be highly diagnostic and allows us to have precise structural assignments. The assigned dimers of PhgPhg and FF have different intermolecular hydrogen bond patterns—namely, two C=O⋯OH hydrogen bonds for PhgPhg and only one C=O⋯OH for FF, both involving peptide C=O groups. In the PhgPhg dimer, phenyl rings can form aromatic interactions with other molecules upon further clustering. These can lead to the formation of various nanostructures in solution for PhgPhg and FF [12], where stacking interactions play an important role in self-assembly. However, due to the lack of electrostatic and hydrophobic interactions in the current gas-phase study, the direct comparison of nanostructures with the gas-phase dimers presented here is somewhat limited. Our findings for dipeptides were compared with cyclo-FF, and from that, a prediction about the vibrational structure of the larger FFF dimer was made. The FFF dimer is expected to form a more ordered structure than FF, with one C=O⋯OH intermolecular hydrogen bond similar to FF and two more NH⋯C=O intermolecular hydrogen bonds. That structure represents the minimal unit cell of an anti-parallel β-sheet.

## Figures and Tables

**Figure 1 molecules-27-02367-f001:**
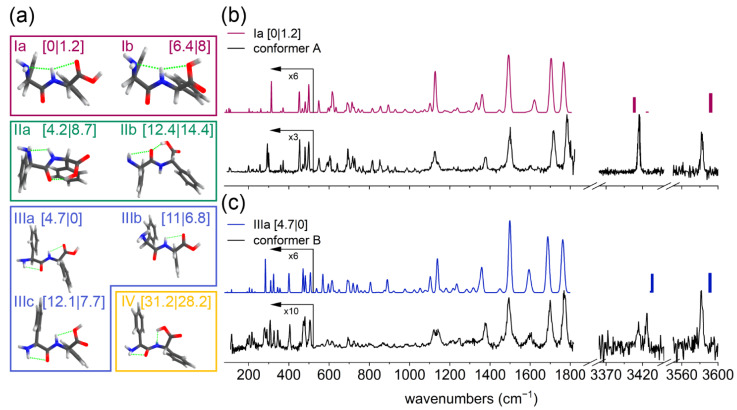
(**a**) The structural families of the PhgPhg monomer with their zero-point energies and Gibbs free energies at 300 K in kJ/mol shown between square brackets; (**b**) experimental IR spectrum of conformer A of the monomer (black) and calculated (pink) IR spectrum of the Ia structure; (**c**) experimental IR spectrum of conformer B of the monomer (black) and calculated (blue) IR spectrum of the IIIa structure. Calculated spectra are scaled by 0.976 (1800–100 cm^−1^) and 0.956 (3370–3600 cm^−1^). The region below 520 cm^−1^ was multiplied for clarity, as indicated.

**Figure 2 molecules-27-02367-f002:**
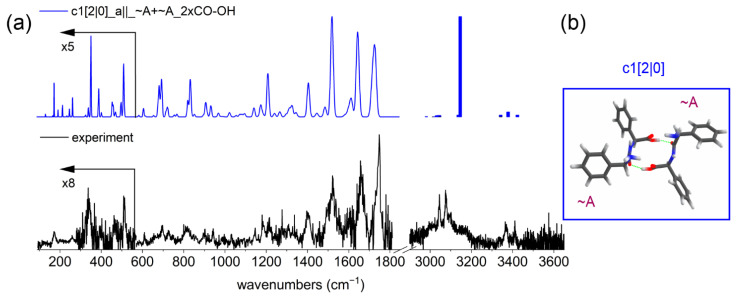
(**a**) Experimental (black, bottom) and the calculated IR spectrum (blue, top) of the dimer of PhgPhg. The calculated spectrum is scaled by 0.976 (1800–100 cm^−1^) and 0.956 (3000–3600 cm^−1^). The region below 570 cm^−1^ was multiplied as indicated for clarity. (**b**) The corresponding structure is shown to the right, which consists of two slightly altered monomeric conformer A structures. The zero-point energy and Gibbs free energy at 300 K in kJ/mol are shown between square brackets.

**Figure 3 molecules-27-02367-f003:**
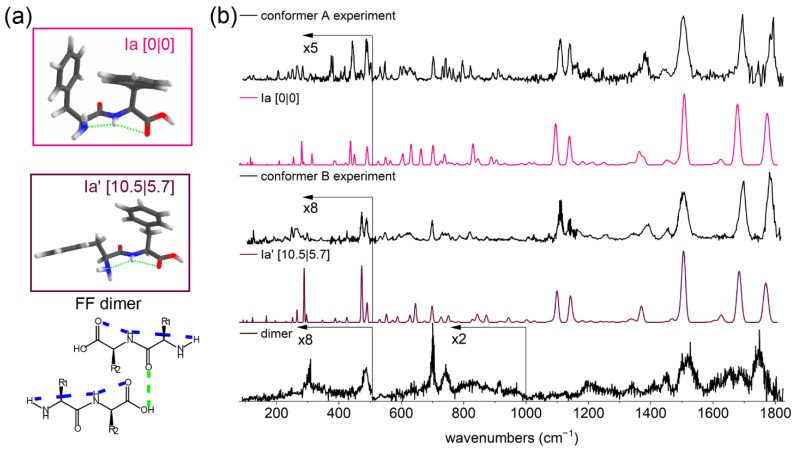
Structural assignment of monomeric FF and IR spectra of the FF monomers and dimer. (**a**) The assigned structures of the FF monomer for conformations A and B with their zero-point energies and Gibbs free energies at 300 K in kJ/mol shown between square brackets. The predicted assigned structure of the FF dimer is shown below with intra- and intermolecular hydrogen bonds in blue and green dashed lines, respectively. Side groups are shown as R_1_ and R_2_. (**b**) The experimental spectra of conformer A and B of the monomer (black) and the calculated IR spectra of the corresponding assigned structures for the monomers (pink and purple) and the experimental IR spectrum of the FF dimer (black, bottom). Calculated spectra are scaled by 0.976. The region below 520 and 1000 cm^−1^ was multiplied for clarity, as indicated in the figure.

**Figure 4 molecules-27-02367-f004:**
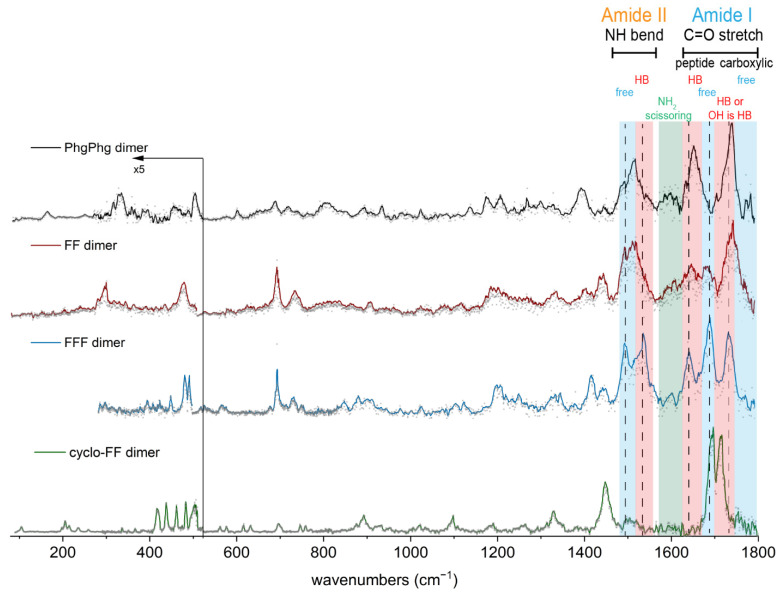
The experimental IR spectra of PhgPhg, FF, FFF, and cyclo-FF dimers (black, red, blue, and green traces, respectively). Grey dots are the averaged data, while solid lines show smoothed data that were obtained using the adjacent-averaging method in Origin software. The far-IR region below about 520 cm^−1^ was multiplied by 5 for clarity. The molecular vibrations in the amide I and II regions are shown with blue (free) and red (intermolecular hydrogen bond—HB) boxes. The black dashed lines show the positions of the vibrations of the FFF dimer.

## Data Availability

All xyz files of the assigned structures can be found in Appendix A, Section 12. The input lines that were used in Gaussian 16 are also listed. Please contact the corresponding author for additional information regarding experimental data.

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
