# Peer review of "Structural Properties of Phenylalanine-Based Dimers Revealed Using IR Action Spectroscopy"

_molecules, 2022, doi:10.3390/molecules27072367_

Round 1
Reviewer 1 Report
This s a very thorough and careful study of gas-phase conformations of small model peptide dimers.
I have just a few questions:
(1) Given the similarities between the calculated IR spectra for different structural families, can the authors comment on how unique the assigment is?
(b) Given the very broad dimer REMPI spectra, how certain are the unique dimer structural assignments? Can mixtures of structures be involved?
Author Response
Attached you will find the answers to all review comments.
We kndly would like to acknowlegde all reviewers for carefully reading our manuscript and provinding with usefull input.
Kind regards,
Anouk Rijs

Reviewer 2 Report
The paper by Stroganova et al. aims to document the very first step of the aggregation of natural peptides based on phenylalanine-related residues, known to form solid nanostructures in aqueous solution. The work capitalizes on the group’s know-how to elucidate the interactions at play in isolated systems, namely a monomer and a dimer, using the powerful sophisticated techniques of the gas phase, namely laser spectroscopies. It is a follow-up of previous works using the same dual experimental and theoretical strategy. Three systems were investigated: monomer and dimers of natural dipeptides of phenylglycine (Phg-Phg) , of phenylalanine (PhePhe) and of the tripeptide of phenylalanine (PhePhePhe).
Sound conclusions are obtained for the PhgPhg monomer, where two conformers are detected, one of them being a very minor one, and the PhgPhg dimer, where no evidence for the existence of minor conformers exists (due to the absence of hole-burning spectra). In both cases the conformers observed match well the most stable structures at 300 K, for both the monomer and the dimer. The assignments are well justified and discussed in terms of IR spectroscopy in all the spectral ranges addressed. Extended monomers are formed, with a C5 interaction. The dimer is made of two monomers in a head-to-tail position, bridged by two antiparallel OH···OC H-bonds connecting the carboxylic groups of each molecule.
The PhePhe system is also informative for the monomer : despite this work actually revisit the spectroscopy of this species, one of the previous works is reassigned and the results are completed by the evidence for the existence of a secondary conformer, convincingly assigned to a closely related species, differing in the orientation of one of the Phe side chains. Regarding the dimer, despite lots of theoretical efforts, no precise assignment could be proposed, but the spectroscopy suggests a unique H-bond bridging two monomer, in which the main conformation is retained.
The paper ends up with a discussion comprising the PhgPhg and PhePhe data together with the structure of the PhePhePhe dimer, the latter being qualitatively assigned from general spectroscopic considerations.
The standard procedures of the gas phase studies have been carried out and obeyed, leaving sound conclusions about the assignment of the IR spectra to well-defined species in the case of small monomers and dimers. The assignments are well justified, thanks to the calculations which are presented. Here the efforts deployed must be acknowledged, owing to the dramatically increasing conformational diversity owing to the size and complexity of these molecules.
The paper is well-written, easy to read despite the complexity of the systems treated. A good balance is achieved between the main text and the details/additional data provided in the SI.
In my opinion, the paper fits the publication standard of the “Molecules” journal; I recommend publication, provided that certain points, listed below, are addressed or clarified by the authors :
- My major concern is the relevance of the specific models studied to the aggregation issues, which the authors pretend to target. For example, most of the applications take place in water, i.e., with at least one charged terminus, and two in the neutral range pH (zwitterionic state). This casts some doubts about the relevance of such isolated conditions, where the molecules are neutral, but with neutral termini. Nanostructures are also observed with capped dipeptides, but in this case the natural neutral termini are also not relevant. Capped models should then be used instead, such as those whose dimers have been described by Gerhards group (e.g., Fricke et al. 2008). Finally in their conclusion, the authors refer to stacking interactions which play an important role in self-assembly. Although these might participate in the driving force for self-assembly, in an aqueous solution hydrophobic effects are probably the major component, which cannot be described nor modelled by the present gas phase study. In my opinion, the relevance of this study of model systems to aggregation issues is somewhat oversold. To avoid misleading the reader, authors should clarify these points, both in the introduction and in the conclusion.
- Chirality is a major issue in these systems and probably also for their applications. No indication is given on the L- or D- conformation of the Phe or Phg residues used : as far as I understand the dipeptides (and the tripeptide) are homochiral (L) : this point is important and should be made clear.
- PhePhe dimer : The spectral congestion of the UV spectrum together with that of the IR bands also suggest that a minor conformation could be present. This point could be mentioned/clarified.
- No PhePhePhe UV spectrum is shown. It would be interesting, in particular, to figure out if it exhibits narrow features, and if this can be connected to the narrow lines of the IR spectrum (narrower than thos- of the PhgPhg and PhePhe dimers). The authors could comment on that.
- Reference to earlier works on similar or related systems could also be made (GlyTrp and TrpGly by Kleinermanns et al.) and the difference with the present species, with a large aromatic content, be discussed. Same remark for the dimers of capped peptides by Gerhards et al. .
- PhgPhg monomer : Fig. S1 : the second red asterisk is missing (the one corresponding to the second probe frequency given in the main text).
- PhePhe dimer : Fig. S5 b) : the HB spectrum shows a unique major band, despite two band systems are anticipated (one for each phenyl chromophore) : Can the authors comment on that ? In the same figure, this band seems to correspond to a weak band (dashed red line) in the top UV spectrum: the B label does not seem to be correctly positioned, since it does not coincide with the dashed line, but rather to the stronger band to the blue.
- Phg : not defined
- Section 3.1 : IR spectra are discussed before the UV features they are based on are presented. Inverting the presentation of the UV and IR/UV spectra would help the reader.
- p. 4 l. 179 rather “roughly T-shaped” than “pi-stacked”
- P5 l 185 : the rotation is not about the peptide bond but about the Calpha(1) - COpep bond.
- The authors seem to anticipate that the population observed is described by a 300 K temperature. The authors should comment on that, especially in relationship with the cold temperatures of the supersonic expansion.
- p. 8 about the monomeric forms conserved in the dimer : There are counterexamples : eg. the capped dipeptides of Gerhards et al., which are beta turns as monomers and beta-strands as dimers.
- P9 l. 395 : dimer is a symmetric molecule : Shouldn’t it be “centro- symmetric” ?
- P 11 l 487 “form” should read “from”
Author Response

(The authors gave the same response as above.)

Reviewer 3 Report
The authors report here an interesting study of structural characterization of monomer and early dimer species during the self-assembly of an important class of peptide-based assembling building blocks, FF and its Phg-Phg variant. The general difficulty in deciphering the processes like this was overcome here by using an approach combining spectroscopic information from mass- and conformer-selective IR method and structural and energetic information from quantum chemical calculations. This information could be valuable for understanding peptide self-assembly and thereby rational design of materials of this kind. Therefore, I think that this paper can be publishable as long as my following comments have been addressed properly.
1) The major goal of the study, as pointed out by the authors, is to examine if the structures of early aggregates like dimers can be extrapolated to those of supramolecular aggregates. It thus would be more informative to compare the structures of monomers and dimers with those observed in reported supramolecular structures, such as two different supramolecular structures of FF reported in previous crystallography studies (Chem Comm 2006, 22, 2332; Angew. Chem. Int. Ed. 2021, 60, e202113845). A comparison between these structures using structural metrics like RMSD is desirable.
2) It is intriguing that there are distinct orientational arrangement of two sidechains of Phg-Phg in its two conformers I and III. In fact, the relative orientation of the two aromatic sides has been suggested as a key parameter defining geometry of FF building blocks (Nanoscale 2014, 6, 2800; J. Phys. Chem. Lett. 2011, 2, 2380; Chem. - Eur. J. 2001, 7, 5153). A computational study has further shown the possible correlation between this parameter and the resulting packing mode of FF in assemblies (ACS Nano 2019, 13, 4455). Can the authors comment on these studies in light of their discovery?
3) As shown in Fig. 1a, there are eight structures that are divided into four groups (I-IV). How were these structures grouped?
4) The major conformer A observed experimentally appears higher in free energy than the minor conformer B (1.2 vs 0.0 kJ/mol) based on quantum chemical calculations. Can the authors comment on this?
Author Response

(The authors gave the same response as above.)
